# Gastrodin Mitigates Ketamine-Induced Inhibition of F-Actin Remodeling and Cell Migration by Regulating the Rho Signaling Pathway

**DOI:** 10.3390/biomedicines13030649

**Published:** 2025-03-06

**Authors:** Ping-Cheng Shih, I-Shiang Tzeng, Yi-Chyan Chen, Mao-Liang Chen

**Affiliations:** 1Department of Anesthesiology, Taipei Tzu Chi Hospital, Buddhist Tzu Chi Medical Foundation, New Taipei City 231016, Taiwan; benego@gmail.com; 2Department of Research, Taipei Tzu Chi Hospital, Buddhist Tzu Chi Medical Foundation, New Taipei City 231016, Taiwan; istzeng@gmail.com; 3Department of Psychiatry, Taipei Tzu Chi Hospital, Buddhist Tzu Chi Medical Foundation, New Taipei City 231016, Taiwan; yichyanc@gmail.com

**Keywords:** *Gastrodia elata*, gastrodin, ketamine, Rho signaling, F-actin, cell migration

## Abstract

**Background/Objects**: Rho signaling plays a role in calcium-regulated cytoskeletal reorganization and cell movement, processes linked to neuronal function and cancer metastasis. *Gastrodia elata*, a traditional herbal medicine, can regulate glutamate-induced calcium influx in PC12 cells and influence cell function by modulating neuronal cytoskeleton remodeling via the monoaminergic system and Rho signaling. This study investigates the effects of gastrodin, a key component of *Gastrodia elata*, on Rho signaling, cytoskeleton remodeling, and cell migration in B35 and C6 cells. It also explores gastrodin’s impact on Rho signaling in the prefrontal cortex of Sprague Dawley rats. **Methods**: B35 cells, C6 cells, and Sprague Dawley rats were treated with ketamine, gastrodin, or both. The expression of examined proteins from B35 cells, C6 cells, and the prefrontal cortex of Sprague Dawley rats were analyzed using immunoblotting. Immunofluorescent staining was applied to detect the phosphorylation of RhoGDI1. F-actin was stained using phalloidin-488 staining. Cell migration was analyzed using the Transwell and wound-healing assays. **Results**: Gastrodin reversed the ketamine-induced regulation of cell mobility inhibition, F-actin condensation, and Rho signaling modulation including Rho GDP dissociation inhibitor 1 (RhoGDI1); the Rho family protein (Ras homolog family member A (RhoA); cell division control protein 42 homolog (CDC42); Ras-related C3 botulinum toxin substrate 1(Rac1)); rho-associated, coiled-coil-containing protein kinase 1 (ROCK1); neural Wiskott–Aldrich syndrome protein (NWASP); myosin light chain 2 (MLC2); profilin1 (PFN1); and cofilin-1 (CFL1) in B35 and C6 cells. Similar modulations on Rho signaling were also observed in the prefrontal cortex of rats. **Conclusions**: Our findings show that gastrodin counteracts ketamine-induced disruptions in Rho signaling, cytoskeletal dynamics, and cell migration by regulating key components like RhoGDI1, ROCK1, MLC2, PFN1, and CFL1. This suggests the potential of gastrodin as a comprehensive regulator of cellular signaling.

## 1. Introduction

Ketamine is an antagonist of the NMDA (N-methyl-d-aspartate) receptor, which is known to induce various psychotic symptoms, and it is used to induce the psychosis model to study metal disorders [1,2,3,4]. It has been reported that changes in calcium homeostasis via the ketamine-induced compensatory upregulation of NMDA receptors lead to cell death via ketamine-induced neuronal apoptosis, ROS generation, and other neurotoxicities [5,6,7,8]. Similarly, ketamine was shown to influence calcium levels in various brain regions in a rat model of psychosis [9]. Additionally, ketamine increased RhoA and ROCK1 expression in cultured hippocampal neurons and induced dendritic spines morphological regulation in hippocampal neurons by modulating the RhoA/ROCK signaling pathway [10].

Calcium levels influence Rho signaling, a pathway critical for treating mental diseases like schizophrenia, depression, post-traumatic stress disorder, Parkinson’s disease (PD), Alzheimer’s disease (AD), Huntington’s disease (HD), and amyotrophic lateral sclerosis [11,12,13]. The Rho signaling pathway can regulate actin–myosin cytoskeleton remodeling that is important for neuronal growth, dendritic spine formation, and also axonal elongation and guidance. Various studies propose that RhoA, Rac1, and Cdc42 modulate the processes of axonal and dendritic spine organization that can regulate neuronal morphology [14,15,16,17]. Dysregulation in RhoA/ROCK1 signaling has been linked to PD, amyloid-β aggregation, tau phosphorylation, synapse injury, and neuroinflammation in AD [18,19,20,21,22,23]. It is also implicated in schizophrenia pathogenesis through proteins like KALRN and other RhoA-associated GAPs and GEFs [24,25,26,27].

*Gastrodia elata* (GE) is a type of orchid native to Asia and is believed to have sedative properties, anti-depression abilities, anti-inflammatory effects, pain-relieving properties and may be beneficial for conditions involving seizures [28,29,30,31,32]. Various studies have demonstrated the protective effects of gastrodin, a major bioactive constituent in GE, on lead-induced hippocampal neuron synaptic plasticity deficits in rats [33], and gastrodin also helps to protect brain cells and improve cognitive function [34,35,36]. Gastrodin can prevent glutamate-induced calcium influx in PC12 cells [37] and, along with 4-hydroxybenzyl alcohol, has antidepressant effects via monoaminergic system modulation and Rho-signaling-related neuronal remodeling [38,39]. Gastrodin can ease ischemia–reperfusion injury and inflammation by modulating SERCA and PLB expression, as well as reducing intracellular calcium overload [40]. A network pharmacological study also mentioned that gastrodin significantly attenuated both Ca^2+^ influx and excitatory synaptic transmission in cortical neurons [41]. GAS was also found to block the Ca^2+^/CaMKII signaling pathway to reduce autophagic flux dysfunction and improve cognitive impairment in VD [42]. These findings suggest that gastrodin could benefit neurons by regulating the calcium concentration in cells to regulate Rho signaling pathways.

In this study, ketamine was used as a psychotic drug to induce Rho signaling regulation in the B35 and C6 cells. The effects of gastrodin on the ketamine-induced regulation of Rho signaling, cell mobility, and F-actin remodel were then examined to investigate the recovery ability of gastrodin in ketamine-treated cells. The results might show the potential of gastrodin in restoring ketamine-induced changes and its connection with Rho signaling regulation.

## 2. Materials and Methods

### 2.1. Cell Culture and Drug Treatment

B35 neuroblastoma cells were cultured in modified Eagle’s medium (Invitrogen, Life Technologies, Eugene, OR, USA) supplemented with 10% fetal bovine serum (FBS; Invitrogen, Life Technologies), 1 mM sodium pyruvate (Invitrogen, Life Technologies), non-essential amino acids (Invitrogen, Life Technologies), and a penicillin–streptomycin (Pen-Strep) solution (Invitrogen, Life Technologies). C6 glioblastoma cells were cultured in Dulbecco’s Modified Eagle Medium with 4.5 g/L glucose (DMEM; Invitrogen, Life Technologies) supplemented with 10% horse serum (Invitrogen, Life Technologies), 2% FBS, 110 mg/L sodium pyruvate (Invitrogen, Life Technologies), and the Pen-Strep solution (Invitrogen, Life Technologies). Cells were maintained in a 37 °C incubator supplemented with 5% CO_2_. To investigate the effects of gastrodin on ketamine-treated cells, B35 and C6 cells were exposed to ketamine for two weeks, followed by a two-week treatment with a combination of ketamine (Ketalar 50 mg/mL injection, Pfizer Limited Co., Sandwich, UK) and gastrodin (Thermo Scientific, Rockford, IL, USA). To assess the independent effects of ketamine or gastrodin, B35 and C6 cells were treated with either ketamine or gastrodin for four weeks. Ketamine was prepared as a stock solution of 50 mM and stored in a refrigerator at 4 °C until use. Gastrodin was dissolved in dimethyl sulfoxide (DMSO) as a stock solution of 100 mM and stored in a refrigerator at −20 °C until use. The final concentrations of ketamine and gastrodin used in all treatments were 200 µM and 10 µM, respectively. The drug concentrations were determined according to our previous publication and the referenced articles [10,43].

### 2.2. Total Protein Extraction and Western Blot Analysis

Proteins from drug-treated B35 and C6 cells were extracted using the CelLytic™ M cell lysis reagent (Sigma-Aldrich, St. Louis, MO, USA) with protease and phosphatase inhibitors (Thermo Scientific) to prevent degradation. Protein samples (10–50 μg) were separated using SDS-PAGE (8%, 10%, or 12.5%) and transferred to PVDF membranes. Membranes were blocked and probed with primary antibodies specific to target proteins, followed by HRP-conjugated secondary antibodies. β-actin was used as an internal control. Protein bands were visualized using the Western Lightning^®^ Plus-ECL kit (PerkinElmer, Inc., Waltham, MA, USA). Primary antibodies used included those for beta-actin (GTX26276, GeneTex Inc., Irvine, CA, USA), RhoGDI1 (#2564, Cell Signaling Technology, Danvers, MA, USA), phosphorylated RhoGDI1 (phospho S174, ab74142, Abcam, Cambridge, UK), RhoA (#2117, Cell Signaling Technology), anti-CDC42 (#2462, Cell Signaling Technology), CDC42 (#2462, Cell Signaling Technology), Rac1 (GTX100761, GeneTex Inc.), phosphorylated RhoA Ser188 (sc-32954, Santa Cruz Biotechnology, Santa Cruz, CA, USA), phosphorylated CDC42 (ab74142, Abcam, Cambridge, UK), phosphorylated Rac1 (phospho S71) (PAB7743, Abnova Corporation, Taipei, Taiwan), ROCK1 (ab45171, Abcam), myosin light chain 2 (#3672, Cell Signaling Technology), phosphorylated MLC2 (Ser-18) (TA309976, OriGene Technologies, Rockville, MD, USA), cofilin-1(H00001072-M04, Abnova Corporation, Taipei, Taiwan), and profilin-1(#3237, Cell Signaling Technology).

### 2.3. Cell Migration and Wound-Healing Assays

In both cell migration and wound-healing assays, B35 and C6 cells treated with 1 × phosphate-buffered saline (PBS) served as the control group. To assess the migration ability of B35 and C6 cells, cells were pretreated with ketamine or gastrodin for seven days, followed by an additional six days of treatment with ketamine, gastrodin, or a combination of both. After the treatment period, 1 × 10^4^ B35 cells or 5 × 10^3^ C6 cells were seeded into Transwell inserts (pore size: 8 μm; Costar, Corning Incorporation, Kennebunk, ME, USA) and cultured for 24 h under continuous stimulation with ketamine, gastrodin, or their combination. Migrated cells were washed with 1 × PBS, fixed with methanol, and stained with a 50 μg/mL propidium iodide solution (Sigma, St. Louis, MO, USA) for 30 min. The stained cells on the membrane were counted under a microscope at 40× magnification. All experiments were performed in triplicate for statistical analysis. The wound-healing assay was performed using drug-treated cells cultured with silicon culture inserts (ibidi, Gräfelfing, Germany) in 12-well culture dishes. After removing the culture inserts, the width of the cell-free gaps was measured at 0, 12, 18, and 24 h. The recovery rate was defined as the percentage reduction in the width of the cell-free gap at each time point relative to the initial gap width at 0 h.

### 2.4. Fluorescent Staining of Active RhoGDI1 and F-Actin in B35 and C6 Cells

B35 and C6 cells were pretreated with ketamine or gastrodin for 7 days, followed by an additional 5 days of treatment with ketamine, gastrodin, or a combination of both. The drug-treated B35 and C6 cells were then seeded into 6-well plates containing poly-L-lysine-coated coverslips and cultured with ketamine, gastrodin, or their combination for an additional 2 days. To assess the activation of RhoGDI1, the cells on the coverslips were fixed with 1% paraformaldehyde and permeabilized with methanol. The fixed cells were then incubated with an anti-phosphorylated-RhoGDI1 (anti-p-RhoGDI1) antibody. An Alexa Fluor^®^ 488 donkey anti-rabbit polyclonal antibody (Abcam, Waltham, MA, USA) was used to detect the expression of p-RhoGDI1 on the cell membrane of B35 and C6 cells. To evaluate F-actin reorganization, drug-treated cells on coverslips were permeabilized with methanol and stained for 60 min using the CytoPainter Phalloidin-iFluor 488 reagent (Abcam, Waltham, MA, USA). The coverslips were subsequently mounted using SlowFade™ Diamond Antifade Mountant (Waltham, MA, USA) (with DAPI), sealed on glass slides, and visualized under a fluorescence microscope.

### 2.5. Animal Preparation and Experimental Protocol

All animal experiments and experimental procedures were approved by the Institutional Animal Care and Use Committee (IACUC) of Taipei Tzu Chi Hospital, Buddhist Tzu Chi Medical Foundation (IACUC-110-027). Six-week-old female Sprague Dawley (SD) rats (purchased from National Laboratory Animal Center, Taipei City, Taiwan) weighing 180–210 g were housed in a colony room with a 12 h light/12 h dark cycle. The average temperature in the room was 24 ± 1 °C, and the average humidity was 50%. Food and water were provided ad libitum. In the groups stimulated with ketamine, the rats were intraperitoneally (i.p.) injected with ketamine daily for 28 days. In the groups treated with *P. cocos*, the rats were i.p. injected with gastrodin daily for 28 days. In the groups treated with ketamine plus gastrodin, the rats were i.p. injected with ketamine daily for 14 days and then with ketamine plus gastrodin for another 14 days. The rats in the control group were i.p. injected with ddH_2_O for 28 days. The rats were randomly divided into the control, ketamine, gastrodin, and ketamine + gastrodin groups (n = 3 per group). Then, 24 h after the last treatment, the rats were sacrificed via CO_2_ asphyxiation, and the prefrontal cortex (PFC) was quickly dissected, frozen in liquid nitrogen for 2 h, and then stored at −80 °C until use. The rats were treated with a final concentration of 50 mg/kg ketamine or/and 10 mg/kg gastrodin [44,45,46].

### 2.6. Statistical Analysis

One-way ANOVA analysis was applied for statistical analysis to compare the differences between the groups (control, ketamine-, ketamine, and gastrodin) in Western blot, the cell migration assay, and the wound-healing assay in this study. *p*-values of <0.05 (*) and <0.01 (**) were defined as statistically significant.

## 3. Results

### 3.1. Gastrodin Regulates the Expression and the Activation of RhoGDI1

To investigate the effects of ketamine and gastrodin on the Rho signaling pathway, the expression and activation of RhoGDI1 in B35 and C6 cells were analyzed. The ketamine treatment increased RhoGDI1 expression levels in both B35 and C6 cells, whereas the gastrodin treatment reduced RhoGDI1 expression (Figure 1a,b). Furthermore, gastrodin mitigated the ketamine-induced upregulation of RhoGDI1 expression in both cell lines (Figure 1a,b). The activation of Rho family proteins is dependent on phosphorylated RhoGDI1 (p-RhoGDI1), the active form of RhoGDI1, which localizes to the inner cell membrane. Fluorescent staining was employed to assess p-RhoGDI1 expression in B35 and C6 cells. As shown in Figure 1c,d, the ketamine treatment reduced the membrane-associated expression of p-RhoGDI1 in both cell lines. The gastrodin treatment, however, preserved p-RhoGDI1 expression levels and reversed the ketamine-induced reduction in p-RhoGDI1 expression in both B35 and C6 cells.

### 3.2. Effect of Gastrodin on Ketamine-Induced Regulation of Rho Family Proteins, RhoA, CDC42, and Rac1

Ketamine has been reported to increase RhoA and ROCK1 expression in the hippocampal neuron of rats [10]. We found that ketamine could also increase RhoA expression in the B35 cells and the C6 cells (Figure 2). Ketamine-induced RhoA expression in B35 cells could be significantly reduced by gastrodin (Figure 2a,c). Gastrodin did not efficiently recover the ketamine-induced expression of RhoA in the C6 cells (Figure 2b,d). CDC42 expressions in the B35 cells were not significantly affected by ketamine and gastrodin (Figure 2a,c). We observed that ketamine reduced CDC42 expression, and gastrodin could recover the ketamine-induced reduction in CDC42 in the C6 cells (Figure 2b,d). We also observed that Rac1 expression could not be affected by either ketamine or gastrodin but would be increased by ketamine combined with gastrodin in the B35 cells (Figure 2a,c). A reduction in Rac1 expression, caused by ketamine in the C6 cells, could be reversed by gastrodin (Figure 2b,d).

### 3.3. Effect of Gastrodin on Ketamine-Induced Activation of Rho Family Proteins, RhoA, CDC42, and Rac1

The GDP form of the Rho family protein can be activated to the phosphorylated GTP form of Rho proteins (p-RhoA, p-CDC42, and p-Rac1) to modulate Rho signaling and downstream biological processes. In Figure 3, we found that ketamine could modulate RhoA phosphorylation to increase the p-RhoA expression level in both B35 and C6 cells. The gastrodin treatment recovered ketamine-induced p-RhoA in both B35 and C6 cells. In contrary, the expressions of p-CDC42 and p-Rac1 were reduced under the ketamine treatment, which were further reversed by the gastrodin treatment in both B35 and C6 cells.

### 3.4. Effect of Gastrdin on Ketamine-Induced Modulation of RhoA/ROCK1/MLC2 Signaling

The activation of Rho family proteins triggers the regulation of Rho signaling. The activation of RhoA induces ROCK1 activity to further modulate MLC2 phosphorylation, which is closely related to actin filament polymerization. We investigate the expression of ROCK1, MLC2, and MLC2 phosphorylation (p-MLC2) in the RhoA/ROCK1 pathway. ROCK1 expression in the B35 cell was reduced by ketamine and was restored by gastrodin (Figure 4a,c). The cleaved (activated) form of ROCK1 was increased by ketamine and was reverted by gastrodin. In C6 cells, ketamine did not affect ROCK1 expression but increased activated ROCK1 expression (Figure 4b,d), whereas the gastrodin treatment restored the ketamine-induced activation of ROCK1 in C6 cells.

In B35 cells, MLC2 phosphorylation was increased under the ketamine treatment, which will decrease under the gastrodin treatment (Figure 4a,c). The phosphorylation of MLC2 would not be affected by either ketamine or gastrodin in C6 cells (Figure 4b,d).

### 3.5. Gastrodin Modulates Ketamine-Modulated NWASP, PFN1, and CFL1 Expression

RhoA could modulate ROCK1, MLC2, formins, and LIMK to further control filament branching and the polymerization of cells through PFN1 and CFL1. CDC42 and Rac1 could also modulate NWASP, Arp2/3, and LIMK to modulate PFN1 and CFL1. We found that ketamine could reduce the expression of NWASP in B35 cells (Figure 5a,c) and could increase NWASP expression in C6 cells (Figure 5b,d). We also revealed that CFL1 and PFN1 would be induced by ketamine and would be restored by the subsequent gastrodin treatment (Figure 5) in both B35 and C6 cells.

### 3.6. Gastrodin Reverses Ketamine-Induced Inhibition of B35 and C6 Cell Migration

By using the wound-healing assay (Figure 6a–c), we found that ketamine inhibited the cell migration ability (recovery of the wound gap area) of B35 and C6 cells. Gastrodin retained the cell migration ability of B35 and C6 cells when compared to the control group. Gastrodin could recover the ketamine-induced inhibition of the cell migration ability of B35 and C6 cells. We further used the Transwell assay and the wound-healing assay to investigate the effects of ketamine and gastrodin on cell migration (Figure 6d). We found that ketamine reduced the cell mobility of B35 and C6 cells. Gastrodin did not significantly affect the cell migration of B35 and C6 cells but would restore the inhibitory effect of ketamine on the cell migration of B5 and C6 cells.

### 3.7. Gastrodin Increases the Levels of F-Actin Nucleation and Polymerization in B35 and C6 Cells

We further examined the reorganization of F-actin using fluorescent staining with phalloidin to reveal the effects of gastrodin on cytoskeleton remodeling in B35 and C6 cells. As shown in Figure 7a, ketamine diminished F-actin nucleation and F-actin filament polymerization in B35 cells, whereas gastrodin induced F-actin nucleation and F-actin filament polymerization in ketamine-treated B35 cells. We also found that ketamine eliminated F-actin filament polymerization in C6 cells, and gastrodin restored it in ketamine-treated C6 cells (Figure 7b). We also observed the effects of ketamine and gastrodin on actin condensation in the cell nuclei of B35 and C6 cells. We showed that ketamine prevented F-actin condensation in the nuclei of B35 (Figure 7c) and C6 (Figure 7d) cells. Gastrodin recovered the diminished effect of ketamine on nuclear actin condensation in B35 and C6 cells.

### 3.8. Effects of Ketamine and Gastrodin on Rho Signaling Regulation in the PFC of SD Rats

Rho family protein regulations in the PFC of SD rats treated with ketamine or/and gastrodin were examined to determine the effects of ketamine/gastrodin on Rho signaling modulation. We found that RhoA expression was increased under the ketamine treatment but was restored by the gastrodin treatment in the PFC of SD rats (Figure 8a,c). Gastrodin restored the expression of CDC42, and Rac1 was downregulated in the ketamine treatment (Figure 8a,c). We also revealed greater p-RhoA expression in the PFC treated with ketamine than in the control SD rats, and it would be mitigated by the gastrodin treatment (Figure 8b,d). Ketamine decreased p-CDC42 and p-Rac1 expressions in the PFC of SD rats (Figure 8b,d), and these effects were significantly reversed by gastrodin.

The expression and activation of ROCK1 was increased in the PFC of ketamine-treated SD rats (Figure 9a,c). Gastrodin reversed the ketamine-induced induction in ROCK1 expression and activation in the PFC. Ketamine increased PFN1 expression in the PFC of the rats, whose expression was reversed by gastrodin (Figure 9a,c). We also revealed that both ketamine and gastrodin did not affect MLC2 expression (Figure 9b,d). MLC2 activation was increased by ketamine and was reversed by gastrodin (Figure 9b,d) in the PFC of rats. Ketamine decreased CFL1 expression in the PFC of the rats, whose expression was recovered by gastrodin (Figure 9b,d).

## 4. Discussion

This study highlights the regulatory effects of gastrodin on the Rho signaling pathway, particularly in counteracting ketamine-induced alterations. Gastrodin modulates Rho family proteins and their downstream signaling mechanisms, providing insights into their molecular actions.

Ketamine upregulates RhoGDI1, a key regulator of Rho GTPases, by disrupting its activation and localization [47]. Gastrodin reduces RhoGDI1 expression and restores membrane-associated phosphorylated RhoGDI1 (p-RhoGDI1), which is crucial for Rho protein activation. By stabilizing Rho signaling, gastrodin prevented the ketamine-induced dysregulation of Rho signaling, F-actin remodeling, and the reduction in cell migration.

Rho proteins (RhoA, CDC42, and Rac1) are vital for cytoskeletal dynamics and cell migration [48]. Ketamine increases RhoA expression and phosphorylated RhoA (p-RhoA) while reducing phosphorylated CDC42 (p-CDC42) and Rac1 (p-Rac1), disrupting cytoskeletal regulation. Gastrodin reverses these effects by lowering RhoA expression in B35 cells and restoring CDC42 and Rac1 activation in C6 cells, demonstrating cell-type-specific regulatory actions. These findings emphasize gastrodin’s role in stabilizing Rho signaling and maintaining cellular homeostasis.

The RhoA/ROCK1 pathway is crucial for regulating actomyosin contractility via MLC2 phosphorylation [49]. Ketamine increased activated ROCK1 and phosphorylated MLC2 (p-MLC2) levels in B35 cells, enhancing actin–myosin interactions. Gastrodin reversed these effects, reducing p-MLC2 and restoring ROCK1 expression. However, the RhoA/ROCK1/MLC2 pathway showed limited responsiveness in C6 cells, suggesting cell-type-specific differences. These findings support the study that RhoA signaling effects are condition-dependent [50].

Rho kinase inhibitors are being explored as neuroprotective agents for glaucoma and neurodegenerative diseases [51,52,53]. Ketamine disrupted F-actin polymerization and nuclear condensation in both B35 and C6 cells, impairing cytoskeletal stability. Gastrodin counteracted these disruptions, promoting F-actin polymerization, restoring nuclear actin condensation, and improving cell migration, as shown in the Transwell and wound-healing assays. Gastrodin’s restorative effects on Rho signaling and actin remodeling proteins (e.g., PFN1 and CFL1) [54,55,56] align with its ability to recover cell motility. By reversing ketamine-induced changes to near-control levels, gastrodin shows promise as a therapeutic agent for conditions linked to dysregulated Rho signaling, with fewer drug effects induced by gastrodin.

The activation of Rho signaling relies on the calcium-induced RhoGDI1 activation of cells. Both ketamine and gastrodin could modulate calcium homeostasis to further affect downstream signaling pathways, including Rho signaling. In the present study, ketamine showed the same regulatory trend on RhoGDI1 phosphorylation and downstream Rho family protein activation in B35 and C6 cells. Interestingly, gastrodin showed its potential to reduce RhoA phosphorylation in B35 cells and induce RhoA phosphorylation in C6 cells. We also observed that ketamine induces relatively higher RhoA activation in C6 cells than in B35 cells. These suggested that there are different activations of the Rho GTPase regulation pathway between different types of cells, which causes differential effects in regulating RhoA phosphorylation in B35 and C6 cells. We found that both ketamine and gastrodin induced the differential regulation of ROCK1, MLC2, and NWASP downstream of RhoA, CDC42, and Rac1 in B35 and C6 cells; these observations could be a result of the differential cross-talk between these proteins in different cell types.

Our in vivo findings in the PFC of ketamine-treated SD rats align with in vitro results. Ketamine activated RhoA and ROCK1 while reducing CDC42, Rac1, and their phosphorylated forms, disrupting cytoskeletal dynamics through altered PFN1 and CFL1 expression. Gastrodin treatment reversed these changes, restoring normal Rho protein activity and downstream effectors. Ketamine and gastrodin may regulate PFN1 and CFL1 expression via LIMK and formin, influencing cell migration. The RhoA/ROCK1 pathway can also inhibit CDC42 and Rac1 activation through these mechanisms. PFN1 and CFL1 are critical for neuron plasticity and dendrite spine stabilization. Dephosphorylated CFL1 promotes actin turnover, lamellipodium formation, and cancer metastasis [57,58,59]. Conversely, PFN1 negatively regulates cell migration in various cancers, including osteoblast, breast, and gastric cancer cells [60,61,62]. Studies also link PFN1 to migration regulation in non-small-cell lung carcinoma (NSCLC) and gastric cancer cells [63,64]. The overexpression of CFL1 inhibits NSCLC invasion but promotes metastasis in triple-negative breast cancer and hepatocellular carcinoma [65,66,67].

In actin dynamics, cofilin-1 and profilin-1 play opposing but complementary roles that are crucial for neuronal function. Dysregulation can lead to neurological disorders like Alzheimer’s disease, Parkinson’s disease, ALS, and Fragile X syndrome. Actin polymerization is enhanced by Profilin-1, while actin disassembly is promoted by cofilin-1 [68]. Our in vitro data show increased PFN1 and CFL1 levels in B35 and C6 cells, suggesting that ketamine-induced disturbance of these proteins may cause abnormal biological functions in cells, such as a reduction in mobility and F-actin remodeling. Gastrodin plays a role in restoring the ketamine-induced reduction in mobility and F-actin remodeling, highlighting gastrodin’s potential therapeutic applications in restoring Rho signaling balance and regulating cell migration in neuronal and glial cells.

## 5. Conclusions

This study demonstrates that gastrodin effectively counteracts the ketamine-induced dysregulation of Rho signaling, cytoskeletal dynamics, and cell migration. Our data also suggest potential therapeutic applications of gastrodin in conditions involving Rho signaling dysregulation. The ability of gastrodin to maintain both upstream regulators (RhoGDI1) and downstream effectors (ROCK1, MLC2, PFN1, and CFL1) highlights its potential as a broad regulator of cellular signaling. Future research should explore the molecular mechanisms underlying the differential effects of gastrodin in distinct cell types and its potential therapeutic applications in neuropsychiatric and cancer models.

## Figures and Tables

**Figure 1 biomedicines-13-00649-f001:**
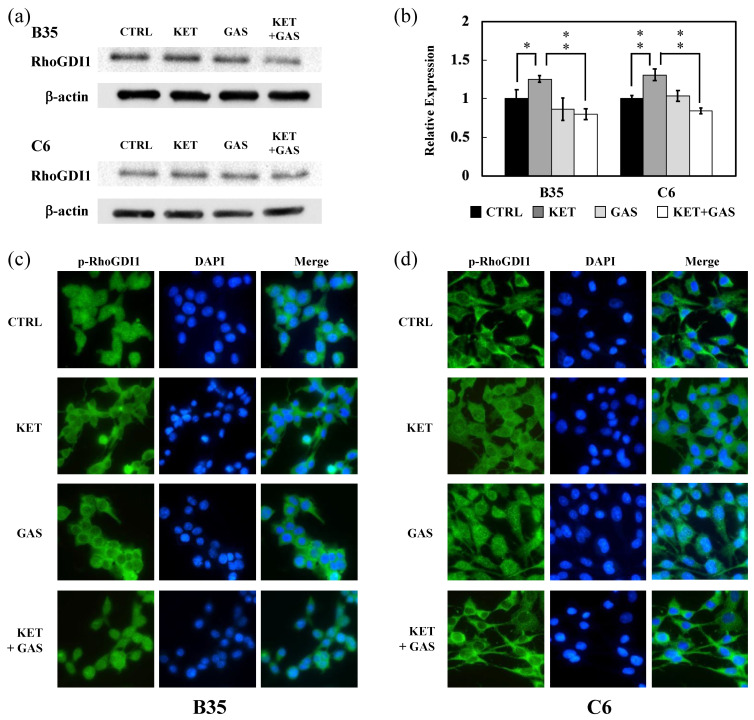
Ketamine and gastrodin regulate the expression and activation of RhoGDI1. Protein extracts collected from B35 and C6 cells treated with ketamine and/or gastrodin were examined for the expression of RhoGDI1 (**a**), and the quantitative results (**b**) are shown. RhoGDI1 phosphorylation was also examined via immunofluorescence microscopy in B35 (**c**) and C6 (**d**) cells. The fluorescent image was captured on a microscope at 20× magnification. The bar charts were generated from triplicate Western blot data from three different batches of drug-treated cells. A *p*-value less than 0.01 (**) or 0.05 (*) from ANOVA followed by Dunnett’s test was considered significant. CTRL, control; ketamine, KET; gastrodin, GAS.

**Figure 2 biomedicines-13-00649-f002:**
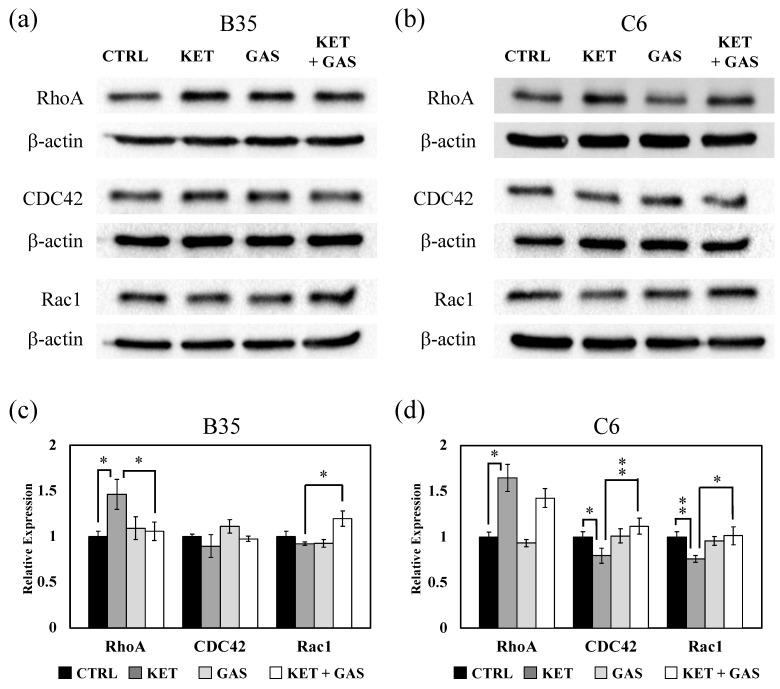
The effects of ketamine and gastrodin on the expression of Rho family proteins. Protein extracted from (**a**) B35 and (**b**) C6 cells treated with ketamine and/or gastrodin were examined for the expression of Rho A, CDC42 and, Rac1. The bar charts (**c**,**d**) of the quantitative results were generated from triplicate Western blot data from three different batches of drug-treated cells. A *p*-value less than 0.01 (**) or 0.05 (*) from ANOVA followed by Dunnett’s test was considered significant. CTRL, Control; ketamine, KET; gastrodin, GAS.

**Figure 3 biomedicines-13-00649-f003:**
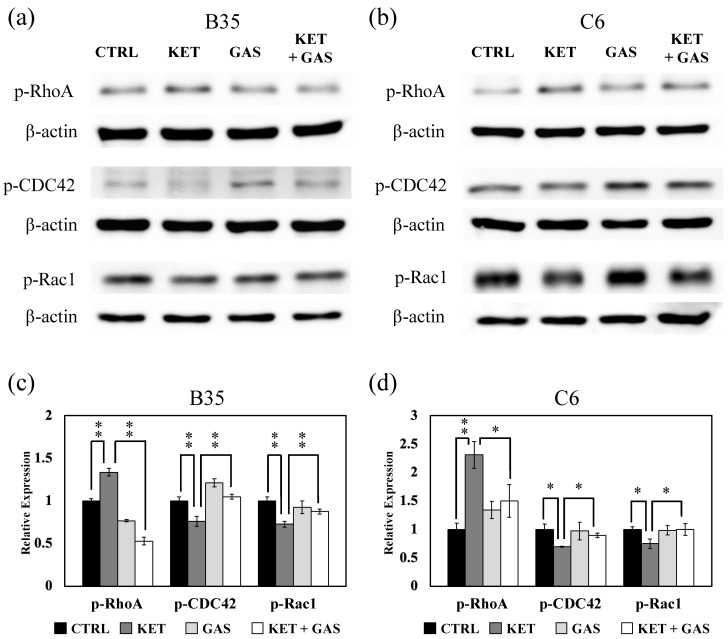
The effects of ketamine and gastrodin on the phosphorylation of Rho family proteins. Protein extracts collected from (**a**) B35 and (**b**) C6 cells treated with ketamine and/or gastrodin were examined for the expression of p-Rho A, p-CDC42, and p-Rac1. The bar charts (**c**,**d**) of the quantitative results were generated from triplicate Western blot data from three different batches of drug-treated cells. A *p*-value less than 0.01 (**) or 0.05 (*) from ANOVA followed by Dunnett’s test was considered significant. CTRL, control; ketamine, KET; gastrodin, GAS.

**Figure 4 biomedicines-13-00649-f004:**
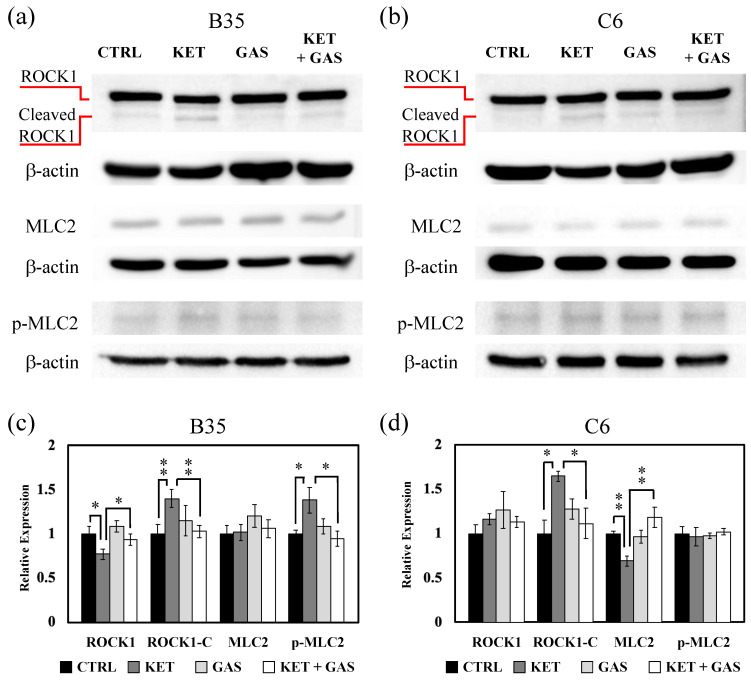
The effects of ketamine and gastrodin on the expression and activation of ROCK1 and MLC2. Protein extracts collected from (**a**) B35 and (**b**) C6 cells treated with ketamine and/or gastrodin were examined for the expression of ROCK1, MLC2, and p-MLC2. The bar charts (**c**,**d**) of the quantitative results were generated from triplicate Western blot data from three different batches of drug-treated cells. A *p*-value less than 0.01 (**) or 0.05 (*) from ANOVA followed by Dunnett’s test was considered significant. CTRL, control; ketamine, KET; gastrodin, GAS.

**Figure 5 biomedicines-13-00649-f005:**
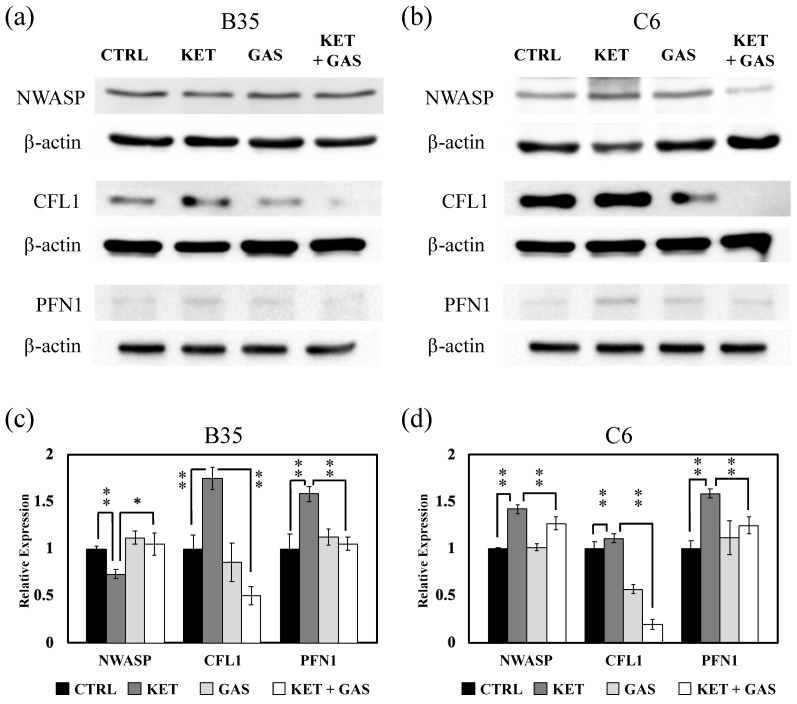
The effects of ketamine and gastrodin on the expression of NWASP, CFL1, and PFN1. Protein extracts collected from (**a**) B35 and (**b**) C6 cells treated with ketamine and/or gastrodin were examined for the expression of NWASP, CFL1, and PFN1. The bar charts (**c**,**d**) of the quantitative results were generated from triplicate Western blot data from three different batches of drug-treated cells. A *p*-value less than 0.01 (**) or 0.05 (*) from ANOVA followed by Dunnett’s test was considered significant. CTRL, control; ketamine, KET; gastrodin, GAS.

**Figure 6 biomedicines-13-00649-f006:**
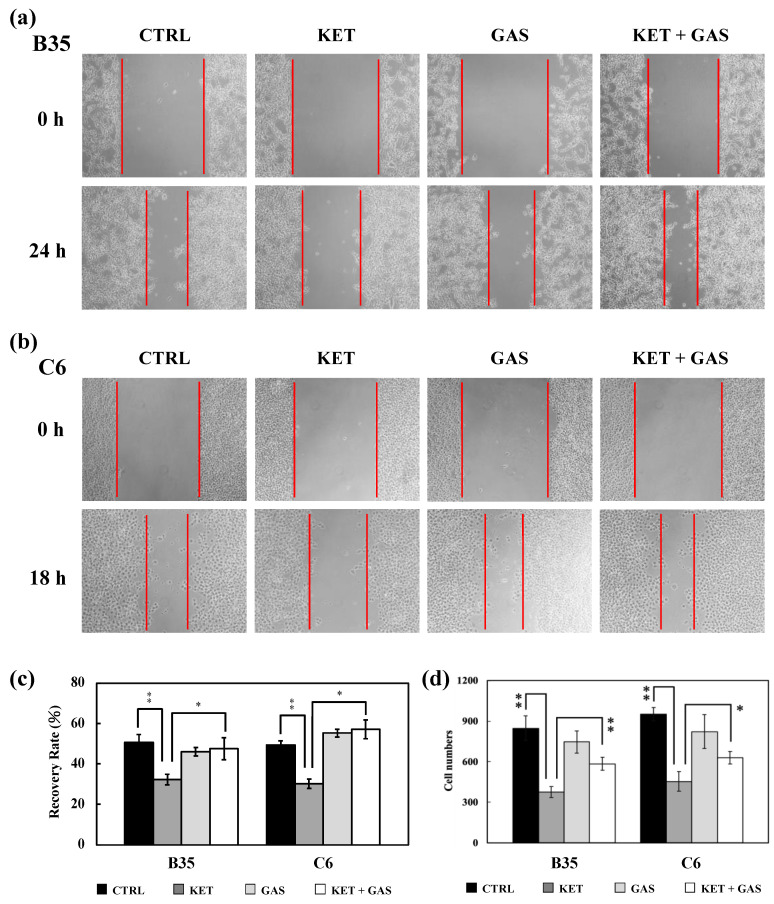
Recovery effect of gastrodin on ketamine-induced cell mobility reduction in B35 and C6 cells. In the wound-healing assay, cell-free gaps in cultured B35 (**a**) and C6 (**b**) cells were observed and recorded at 0 h and 24/18 h. The image was captured on a microscope at 10× magnification. The recovery rate was calculated as the percentage reduction in the width of the cell-free gaps in B35 and C6 cells at 24/18 h compared with the cell-free gap at 0 h (**c**). Cell mobility was measured via the Transwell migration assay and the wound-healing assay. In the Transwell migration assay (**d**), the migrated cells in each group were counted to indicate the cell migration ability. The bar charts of the quantitative results were generated from triplicate assays from three different batches of drug-treated cells. A *p*-value of less than 0.01 (**) or 0.05 (*) was considered significant. CTRL, control; ketamine, KET; gastrodin, GAS.

**Figure 7 biomedicines-13-00649-f007:**
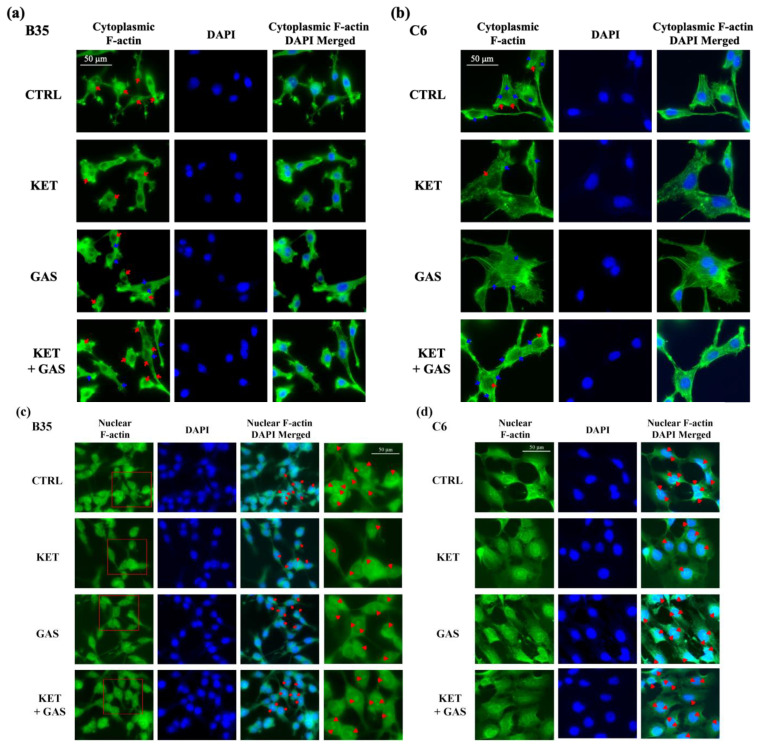
Effects of ketamine and gastrodin on F-actin filament condensation and nucleation in B35 and C6 cells. F-actin filament condensation and nucleation were examined via immunofluorescence staining with phalloidin. Gastrodin reversed the inhibitory effect of ketamine on F-actin nucleation (red arrows) and F-actin filament condensation (blue arrows) in the cytoplasm of (**a**) B35 and (**b**) C6 cells. Gastrodin reversed the inhibitory effect of ketamine on F-actin nucleation (red arrows) in the nuclei of (**c**) B35 and (**d**) C6 cells. The image on the coverslip was captured on a microscope at 40× magnification.

**Figure 8 biomedicines-13-00649-f008:**
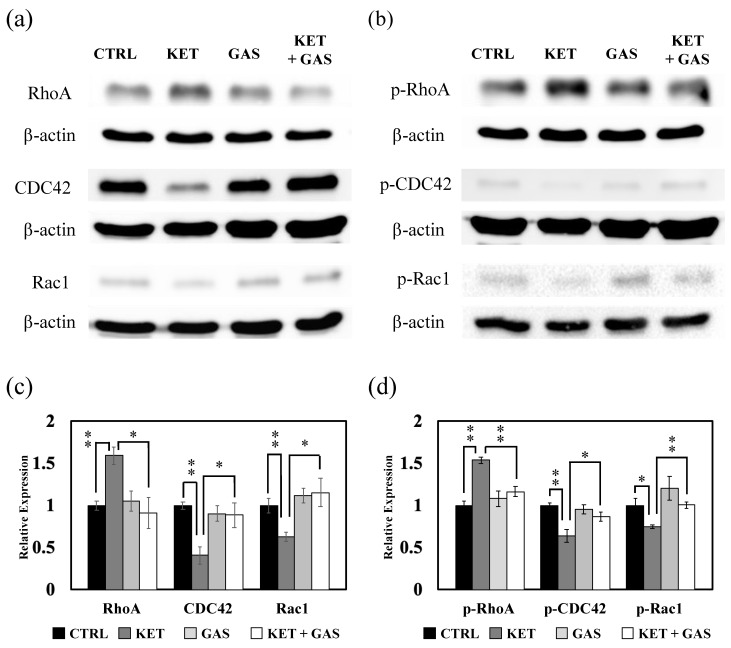
The effects of ketamine and gastrodin on the expression and phosphorylation of RhoA, CDC42, and Rac1 in the PFC of SD rats. Protein extracts collected from the PFC of SD rats treated with ketamine and/or gastrodin were examined for (**a**) the expression and (**b**) phosphorylation of RhoA, CDC42, and Rac1. The bar charts (**c**,**d**) of the quantitative results were generated from triplicate Western blot data from three drug-treated rats. A *p*-value less than 0.01 (**) or 0.05 (*) from ANOVA followed by Dunnett’s test was considered significant. CTRL, control; ketamine, KET; gastrodin, GAS.

**Figure 9 biomedicines-13-00649-f009:**
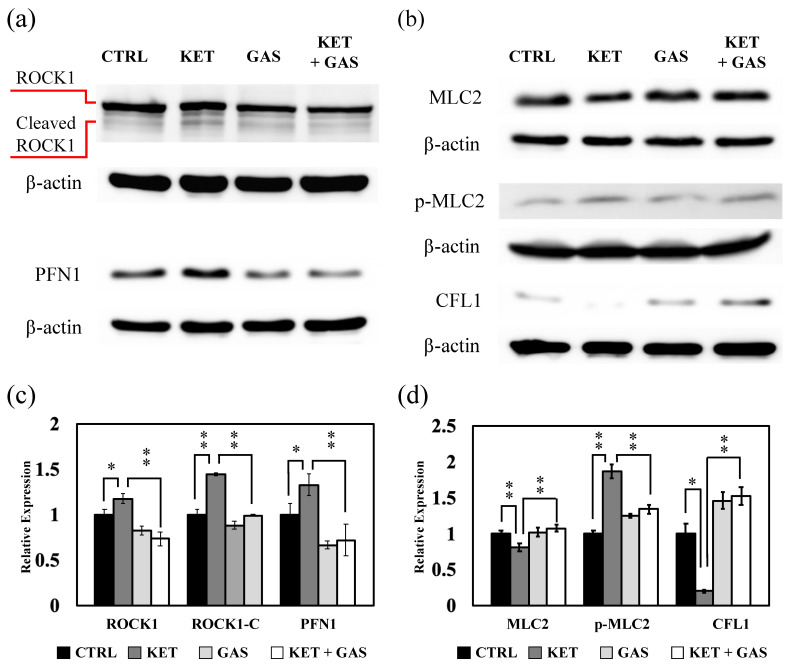
The effects of ketamine and gastrodin on the regulation of ROCK1, MLC2, CFL1, and PFN1 in the PFC of SD rat. Protein extracts collected from the PFC of SD rats treated with ketamine and/or gastrodin were examined for the regulation of (**a**) ROCK1 and PFN1 and (**b**) MLC2 and FCL1. The bar charts (**c**,**d**) of the quantitative results were generated from triplicate Western blot data from three drug-treated rats. A *p*-value less than 0.01 (**) or 0.05 (*) from ANOVA followed by Dunnett’s test was considered significant. CTRL, control; ketamine, KET; gastrodin, GAS.

## Data Availability

The data that support the findings of this study are available from the corresponding author upon reasonable request.

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
