# Peer review of "Gastrodin Mitigates Ketamine-Induced Inhibition of F-Actin Remodeling and Cell Migration by Regulating the Rho Signaling Pathway"

_biomedicines, 2025, doi:10.3390/biomedicines13030649_

Round 1
Reviewer 1 Report
Comments and Suggestions for Authors
- Why the final concentrations of ketamine and gastrodin used in all treatments were 200 μM and 10 μM, for treating B35 and C6 cells? Explain why those concentrations were used?
- Was the wound healing experiment done with cells that were not treated, with controlled cells? I have seen in the figure that control cells were done, but in the text it is not explained. It is not stated anywhere in the text under materials and methods that wound healing was also done on control, untreated cells?
- Everywhere where actin is indicated, as a control, in the images it is necessary to replace the names with β-actin instead of Actin?
Author Response
Reply to the reviewer's comments
Reviewer 1
We thank the reviewer for the valuable comments. We reply to the reviewer's comments point by point as follows
- Why the final concentrations of ketamine and gastrodin used in all treatments were 200 μM and 10 μM, for treating B35 and C6 cells? Explain why those concentrations were used?
Cell migration ability and actin remodeling could be sufficiently induced in B35 and C6 cells treated with 200 μM ketamine according to our previous study. Some studies used 100-400 μM to culture neuronal cells and C6 cells. 200 μM ketamine was then used to perform all the tests in this study according to our previous publication and previous articles. (Ketamine destabilizes growth of dendritic spines in developing hippocampal neurons in vitro via a Rho‑dependent mechanism. https://doi.org/10.3892/mmr.2018.9531; NMDAR-independent, cAMP-dependent antidepressant actions of ketamine. https://doi.org/ 10.1038/s41380-018-0083-8)
The gastrodin concentration used in this study was modified and determined according to the published articles. (Gastrodin Inhibits H2O2-Induced Ferroptosis through Its Antioxidative Effect in Rat Glioma Cell Line C6. https://doi.org/10.1248/bpb.b19-00824; Gastrodin promotes CNS myelination via a lncRNA Gm7237/miR-142a/MRF pathway. https://doi.org/10.1080/15476286.2020.1841976)
These references have been cited in the manuscripts. (Line 100-102)
- Was the wound healing experiment done with cells that were not treated, with controlled cells? I have seen in the figure that control cells were done, but in the text it is not explained. It is not stated anywhere in the text under materials and methods that wound healing was also done on control, untreated cells?
In all the wound healing assay, the cells in the control group were treated with 1×PBS. We will include this description in the Materials and Methods section of the manuscript. (Line 123-124)
- Everywhere where actin is indicated, as a control, in the images it is necessary to replace the names with β-actin instead of Actin?
We have replaced all indications of “Actin” with “b-actin” in all the images.
Reviewer 2 Report
Comments and Suggestions for Authors
The study conducted by Ping-Cheng Shih et al., entitled "Gastrodin Mitigates Ketamine-induced Inhibitory of F-actin Remodeling and Cell Migration by Regulating the Rho Signaling Pathway," highlights the importance of gastrodin in biological functions, with a focus on the Rho signaling pathway. The authors aimed to investigate the potential role of gastrodin in mitigating ketamine-induced effects and its connection to Rho signaling regulation. Based on their key findings, the authors suggest that gastrodin may be effective in treating neuropsychiatric disorders involving dysregulation of Rho signaling. Overall, the study appears to be well-conducted and merits publication, as the authors employed rigorous testing to validate their results. However, I have some concerns that need to be addressed before publication.
The title should be revised to replace "Ketamine-induced inhibitory" with "Ketamine-induced inhibition."
The authors claim that gastrodin is the major component of Gastrodia elata, but they did not mention the source from which they obtained gastrodin. The authors should provide information on whether gastrodin was purchased or isolated and purified from Gastrodia elata, and if so, describe the method used. Furthermore, they should assure the purity of the gastrodin used in the study.
In the methods section, the authors used ketamine at a dose of 50 mg/kg and gastrodin at 10 mg/kg. They should provide a rationale for selecting these specific doses and discuss any potential toxicity of gastrodin in experimental animals.
The authors should explain why they used MK-801 in their study.
There are several grammatical errors throughout the text that require careful attention before publication.
Author Response
Reply to the reviewer's comments
Reviewer 2
We thank the reviewer for the valuable comments. We reply to the reviewer's comments point by point as follows
- The title should be revised to replace "Ketamine-induced inhibitory" with "Ketamine-induced inhibition."
Reply:
The title of the manuscript has been modified according to reviewer’s comment.
- The authors claim that gastrodin is the major component of Gastrodia elata, but they did not mention the source from which they obtained gastrodin. The authors should provide information on whether gastrodin was purchased or isolated and purified from Gastrodia elata, and if so, describe the method used. Furthermore, they should assure the purity of the gastrodin used in the study.
Reply:
The gastrodin was purchased from Thermo Scientific Chemical. It has been addressed in the manuscript. (Line 95)
- In the methods section, the authors used ketamine at a dose of 50 mg/kg and gastrodin at 10 mg/kg. They should provide a rationale for selecting these specific doses and discuss any potential toxicity of gastrodin in experimental animals.
Reply:
The use of ketamine at a dose of 50 mg/kg was based on the sub-anesthetic dose of 25-50 mg/kg, as referenced in previous articles, published by a coauthor listed in this manuscript, that can induce behavioral changes in rats and mice.
Different concentrations of gastrodin (5-300mg/kg) were used to study the neuroprotective effect of gastrodin on animals. Based on some studies with similar research designs, the concentration of gastrodin was selected as 10 mg/kg in this study. The decision also took into account that gastrodin has sufficient neuroprotective effects and a small potential for neurotoxicity.
The referenced articles have been cited in the manuscript.
- The authors should explain why they used MK-801 in their study.
Reply:
Our laboratory is conducting experiments on the association between Poria and MK801 and is preparing a manuscript. The description of MK-801 was mistyped during the preparation of this manuscript and has been corrected to Gastrodin.
- There are several grammatical errors throughout the text that require careful attention before publication.
Reply:
We have checked the grammatical errors throughout the text.
Reviewer 3 Report
Comments and Suggestions for Authors
The research is interesting, but the manuscript raises some questions.
The relevance of the study in the Introduction is unclear. If the problem that the work is aimed at solving is the effects of ketamine, then this is where the introduction should begin. Describe the need and extent of ketamine use, what side effects occur, why they need to be overcome, and not just stop taking ketamine. After that, describe the gastrodin as a way to overcome these undesirable effects and a possible mechanism.
In the Introduction, the authors use mixed GE and gastrodin, which causes misunderstanding. "The pharmacological effects of GE include regulating calcium levels in cells" - is it true? Does the orchid regulate calcium levels?
The methods section is incomplete. It is necessary to describe the source of the substance, concentration, storage conditions before experiments, and so on.
In method section 2.3 the assay with Transwell inserts were described. However, the results of this assay were not reported.
All figures have a very strange look. It looks like the drawings are low resolution, but stretched to a large size.
The title of section 3.2 (and others) "Ketamine and Gastrodin Regulate Expression of Rho Family Protein-RhoA, CDC42, and Rac1." raises questions. What is the purpose of the study? Is the effect of two compounds being investigated, or is the effect of gastrodin on ketamine-treated cells being investigated?
The use of animals in this study raises doubts about the ethics of the experiment. Methods of euthanasia and anesthesia are not described. So many rats to make several immunoblots? This is very irrational and unethical. What is the MK-801 in this experiment?
The authors' conclusion that "There is no evidence that this substance affects the course of neuropsychiatric diseases" is unfounded. No behavioral tests were conducted.Our data also suggests potential therapeutic applications of gastrodin in conditions involving Rho signaling dysregulation, such as neuropsychiatric disorders.
The authors need to carefully review the goals and conclusions.
Comments on the Quality of English LanguageEnglish needs to be checked.
Author Response
Reply to the reviewer's comments
Reviewer 3
We thank the reviewer for the valuable comments. We reply to the reviewer's comments point by point as follows
- The relevance of the study in the Introduction is unclear. If the problem that the work is aimed at solving is the effects of ketamine, then this is where the introduction should begin. Describe the need and extent of ketamine use, what side effects occur, why they need to be overcome, and not just stop taking ketamine. After that, describe the gastrodin as a way to overcome these undesirable effects and a possible mechanism. In the Introduction, the authors use mixed GE and gastrodin, which causes misunderstanding. "The pharmacological effects of GE include regulating calcium levels in cells" - is it true? Does the orchid regulate calcium levels?
Reply:
The introduction section of the manuscript has been thoroughly rewritten and restructured, and we have added more information about the correlation between ketamine, calcium, and rho signaling.
- The methods section is incomplete. It is necessary to describe the source of the substance, concentration, storage conditions before experiments, and so on.
Reply:
We have improved the Methods section and added the appropriate descriptions of substances used in this study.
- In method section 2.3 the assay with Transwell inserts were described. However, the results of this assay were not reported.
Reply:
Figure 6(d) is the result of Transwell migration assay.
- All figures have a very strange look. It looks like the drawings are low resolution, but stretched to a large size.
Reply:
The blurry images have been replaced with a clear vision of images.
- The title of section 3.2 (and others) "Ketamine and Gastrodin Regulate Expression of Rho Family Protein-RhoA, CDC42, and Rac1." raises questions. What is the purpose of the study? Is the effect of two compounds being investigated, or is the effect of gastrodin on ketamine-treated cells being investigated?
Reply:
In this study, ketamine was used as a psychotic drug to induce the Rho signaling regulation in the B35 and C6 cells. The effects of gastrodin on ketamine-induced regulation of Rho signaling, cell mobility, and F-actin remodel were then examined to investigate the recovery ability of gastrodin in ketamine-treated cells.
We have modified the sub-titles that does not clearly present the purposes and goals of the study in the result section. The goal of the study has been raised in the modified “Introduction” section.
- The use of animals in this study raises doubts about the ethics of the experiment. Methods of euthanasia and anesthesia are not described. So many rats to make several immunoblots? This is very irrational and unethical. What is the MK-801 in this experiment?
Reply:
In this study, rats were killed by carbon dioxide asphyxiation. We have added this description into the manuscript. In animal studies, we used only three animals per group and established triplicate experiments with animals as a protein source. The total number of animals used in this study was twelve. Protein extracts were prepared from three animals in each group for detecting all target proteins; not every target protein had its group of three animals.
Our laboratory is conducting experiments on the association between Poria and MK801 and is preparing a manuscript. The description of MK-801 was mistyped during the preparation of this manuscript and has been corrected to Gastrodin.
- The authors' conclusion that "There is no evidence that this substance affects the course of neuropsychiatric diseases" is unfounded. No behavioral tests were conducted. Our data also suggests potential therapeutic applications of gastrodin in conditions involving Rho signaling dysregulation, such as neuropsychiatric disorders.
Reply:
We have deleted the inappropriate connection between the results we found with neuropsychiatric disease. We only concluded the regulatory mechanism of Rho signaling affected by gastrodin and ketamine found in this study.
- The authors need to carefully review the goals and conclusions.
Reply:
We have revised our goals and conclusions to avoid misunderstandings by readers.
Reviewer 4 Report
Comments and Suggestions for Authors
This study suggested that gastrodin counteracts ketamine-induced disruptions in Rho signaling, cytoskeletal remodeling and cell migration in B35 and C6 cells, as well as in the prefrontal cortex of Sprague-Dawley rats. By modulating key signaling components such as RhoGDI1, ROCK1, MLC2, PFN1, and CFL1, gastrodin emerges as a potential regulator of cellular dynamics. Here are some suggestion below:
- In introduction section, author should write more about the ketamine and calcium and rho signaling, adding more evidence increased the understanding of readers.
- In result section, author should change the western image of cleaved ROCK, Rac1, Cdc42, MLC2, PFN1, CFL1 and pMLC2. It is not very clear.
- In figure 7a, b, c and d author should add scale bar and write the magnification in legend. It should enhance result reliability.
- Author mentions that actin remodeling proteins (PFN1, CFL1) but does not clearly connect these molecular changes to functional outcomes, such as neuronal plasticity. Or including similar evidences in the discussion part at preclinical and clinical level.
- The differential effects observed in B35 and C6 cells are intriguing. Expanding the discussion on why these variations occur (e.g., differences in Rho protein expression baselines, receptor profiles, or signaling crosstalk) would improve clarity.
- Author should improve some content such as “gastrodin shows promise as a therapeutic agent for conditions linked to dysregulated Rho signaling, with fewer side effects.” The phrase “fewer side effects” needs clarification—compared to what?
- Overall manuscript is well designed and adding significant insight for future research.
Author Response
Reply to the reviewer's comments
Reviewer 4
We thank the reviewer for the valuable comments. We reply to the reviewer's comments point by point as follows
- In introduction section, author should write more about the ketamine and calcium and rho signaling, adding more evidence increased the understanding of readers.
Reply:
The introduction section of the manuscript has been thoroughly rewritten and restructured, and we have added more information about the correlation between ketamine, calcium, and rho signaling.
- In result section, author should change the western image of cleaved ROCK, Rac1, Cdc42, MLC2, PFN1, CFL1 and pMLC2. It is not very clear.
Reply:
The blurry images have been replaced with a clear vision of images.
- In figure 7a, b, c and d author should add scale bar and write the magnification in legend. It should enhance result reliability.
Reply:
The scale bar and the magnification have been added in the figures and legend.
- Author mentions that actin remodeling proteins (PFN1, CFL1) but does not clearly connect these molecular changes to functional outcomes, such as neuronal plasticity. Or including similar evidences in the discussion part at preclinical and clinical level.
Reply:
We have added some discussion to show the possible connections of disturbance of PFN1 and CFL1 to the physiological function damage observed in this study. We also mentioned some neurological diseases reported that link to the PFN1 and CFL1.
- The differential effects observed in B35 and C6 cells are intriguing. Expanding the discussion on why these variations occur (e.g., differences in Rho protein expression baselines, receptor profiles, or signaling crosstalk) would improve clarity.
Reply:
Activation of Rho signaling relies on the calcium-induced RhoGDI1 activation in cells. Both ketamine and gastrodin could modulate calcium homeostasis to further affect downstream signaling pathways, including Rho signaling. In the present study, ketamine showed the same regulatory trend on RhoGDI1 phosphorylation and downstream Rho family protein activation in B35 and C6 cells. Interestingly, gastrodin showed its potential to reduce RhoA phosphorylation in B35 cells but to induce RhoA phosphorylation in C6 cells. We also observed ketamine induces relatively higher RhoA activation in C6 cells rather than in B35 cells. These suggested that there are different activations of the Rho GTPase regulation pathway between different types of cells which causes differential effects in regulating RhoA phosphorylation in B35 and C6 cells. We investigated that both ketamine and gastrodin induced differential regulation of ROCK1, MLC2, and NWASP regulation downstream the RhoA, CDC42, and Rac1 in B35 and C6 cells, these observations could be a result of differential cross-talk of these proteins between different cell types. These observations have been discussed the discussion section.
- Author should improve some content such as “gastrodin shows promise as a therapeutic agent for conditions linked to dysregulated Rho signaling, with fewer side effects.” The phrase “fewer side effects” needs clarification—compared to what?
Reply:
The efficacy of antipsychotic drugs comes from the regulation of cellular proteins, and these regulations may cause other cell damage, that is, the occurrence of side effects. Gastrodin can alleviate the regulation of Rho signaling, cell movement and cytoskeleton reorganization induced by ketamine, but gastrodin itself has no effect on these changes. Since we did not connect the effects of gastrodin on cells to possible damage caused in cells, we changed the description 'side effects' to 'drug effects' to avoid over-explaining our results.
- Overall manuscript is well designed and adding significant insight for future research.
Reply:
We thank the reviewer for the positive comment.
Round 2
Reviewer 3 Report
Comments and Suggestions for Authors
After revision, the manuscript can be accepted for publication.